# Foreskin surface area is not associated with sub-preputial microbiome composition or penile cytokines

Godfrey Kigozi[1☯], Cindy M. Liu[2☯], Daniel Park[2], Zoe R. Packman[3], Ronald H. Gray[1,4], Rupert Kaul[5], Aaron A. R. Tobian[1,3], Alison G. Abraham[4,6], Joseph Ssekasanvu[1,4], Joseph Kagaayi[1], Jessica L. Prodger[7,8,9]*

1 Rakai Health Sciences Program, Kalisizo, Uganda, 2 Department of Environmental and Occupational Health, Milken Institute School of Public Health, George Washington University, Washington, DC, United States of America, 3 Department of Pathology, Johns Hopkins University School of Medicine, Baltimore, MD, United States of America, 4 Department of Epidemiology, Johns Hopkins Bloomberg School of Public Health, Baltimore, MD, United States of America, 5 Department of Medicine, University of Toronto, Toronto, Ontario, Canada, 6 Department of Ophthalmology, School of Medicine, Johns Hopkins University, Baltimore, MD, United States of America, 7 Department of Microbiology and Immunology, Schulich School of Medicine and Dentistry, Western University, London, Ontario, Canada, 8 Department of Epidemiology and Biostatistics, Schulich School of Medicine and Dentistry, Western University, London, Ontario, Canada, 9 Department of Medicine, School of Medicine, Johns Hopkins University, Baltimore, MD, United States of America

☯ These authors contributed equally to this work.
* jprodge@uwo.ca

**Data Availability Statement:** All relevant data are within the paper and its Supporting Information files.

## Abstract

### Objective

Male circumcision (MC) reduces acquisition of HIV-1 in heterosexual men by at least 60%, but the biological mechanism for this protection is incompletely understood. Previous studies have shown that a larger foreskin size, increased abundance of anaerobic bacteria in the sub-preputial space, and higher levels of pro-inflammatory cytokines on the penis are all prospectively associated with risk of HIV-1 acquisition. Since coverage of the glans on the non-erect penis is dependent on foreskin size, a larger foreskin could result in a less aerobic environment that might preferentially support anaerobic bacterial growth and induce inflammation. We therefore assessed the relationship between foreskin size, penile microbiome composition and local inflammation.

### Methods

This is a retrospective, cross-sectional analysis of 82 HIV-uninfected men who participated in a randomized trial of MC for HIV-1 prevention in Rakai, Uganda between 2003–2006. Sub-preputial swabs were collected prior to MC and assessed for cytokines (multiplexed immunosorbent assay) and bacterial load (qPCR) and taxon abundance (sequencing). Foreskin size was measured immediately after MC.

### Results

Foreskin surface area did not correlate with total bacterial load (rho = 0.05) nor the abundance of key taxa of bacteria previously associated with HIV-1 risk (rho = 0.04–0.25).

**Funding:** This work was supported by grants from the National Institutes of Health (www.nih.gov), grant numbers R01AI128779 (AT) and R01AI123002 (CL). The funders had no role in study design, data collection and analysis, decision to publish, or preparation of the manuscript.

**Competing interests:** The authors have declared that no competing interests exist.

Foreskin surface area also did not correlate with sub-preputial cytokine concentrations previously associated with HIV-1 risk (IL-8 rho = 0.05).

## Conclusions

Larger foreskin size is not associated with either increased penile anaerobes or pro-inflammatory cytokines. These data suggest that foreskin size does not increase HIV-1 risk through changes in penile microbiome composition or penile inflammation.

## Introduction

Three large randomized trials have shown that male circumcision reduces the risk of human immunodeficiency virus-1 (HIV-1) acquisition in heterosexual men by at least 60% [1–3] and also other viral sexually transmitted infections such as herpes simplex virus type two (HSV-2) and human papillomavirus (HPV)[4–9]. However, the biological mechanism by which this protection is conferred remains incompletely understood [10].

We previously reported that on the penis both the density of specific anaerobic bacterial genera and local levels of pro-inflammatory cytokines are associated with increased risk of HIV-1 seroconversion in uncircumcised men [11, 12]. Additionally, the abundance of anaerobic genera on the penis correlates positively with local concentrations of these cytokines [11]. One possible interpretation for these findings is that anaerobic bacteria increase HIV-1 risk by driving penile inflammation. Mucosal inflammation is a demonstrated risk factor for cervico-vaginal HIV-1 acquisition, and is associated with an increased number and relative susceptibility of HIV-1 target cells in the mucosa [13, 14], altered dendritic cell sampling [15], and decreased epithelial barrier function [16, 17].

An epidemiologic study showed that a larger foreskin surface area is also associated with an increased risk of HIV-1 seroconversion [18]. Since coverage of the glans on the non-erect penis is variable and dependent on foreskin size, a deeper foreskin fold from a larger foreskin could result in a less aerobic environment that might preferentially promote anaerobic bacterial growth and induce inflammation. However, it has not yet been explored if foreskin size is associated with penile microbiome composition and local inflammation.

## Materials and methods

This research was conducted with the written consent of all participants. Ethical approval was obtained from the Research and Ethics Committee of the Uganda Virus Research Institute, the Ugandan National Council for Science and Technology, the Committee for Human Research at Johns Hopkins University and Western Institutional Review Board. The trial was registered at Clinical.Trials.Gov NCT00425984.

This is a retrospective, cross-sectional analysis of data collected during a randomized trial of circumcision for HIV prevention, conducted in Rakai, Uganda between 2003 and 2006 [2]. At the time this clinical trial was conducted there was no evidence that circumcision reduced HIV risk, and thus enrolment was performed at a community level without selection for individuals at high risk for HIV acquisition. All men were HIV-uninfected and free from symptomatic genital infections at trial enrollment, and were randomized to either receive circumcision immediately or after 24 months. Among men randomized to receive immediate circumcision, foreskin surface area was evaluated immediately upon removal, by measuring length and width at the midpoint with mild tension applied at all four corners [18]. A coronal

sulcus swab was collected prior to circumcision, resuspended in Amplicor medium (Roche Diagnostics, Indianapolis, USA) and stored at -80˚C.

Coronal sulcus swabs were analyzed for penile microbiome composition and secreted cytokine concentrations as previously described [11, 12]. In brief, cytokines were measured using a custom multiplex kit (Meso Scale Discovery; Rockville, USA) for: IL-1α (interleukin-1α), IL-8, MCP-1 (monocyte chemotactic protein-1), MIG (monokine induced by γ-interferon), MIP-3α, RANTES (Regulated on Activation, Normal T cell Expressed and Secreted), and GM-CSF (granulocyte macrophage colony-stimulating factor). For microbiome characterization, total DNA was extracted from swabs and penile bacterial load was quantified using broad-coverage quantitative PCR (qPCR) targeting the V3-V4 region, measured as 16S rRNA gene copies per swab. Penile microbiome composition was characterized by sequencing the V3V4 region of the bacterial 16S rRNA gene and taxonomic classification performed using a Naïve Bayesian Classifier (v.2.10) [19]. Absolute abundance of each penile bacterial genus was calculated as: penile bacterial load x proportional abundance of the given genus.

As previously reported, penile cytokine concentrations were generally low, with only IL-8 being quantifiable in >50% of men [11]. Therefore, cytokine concentrations were dichotomized (detectable or undetectable) and their associations with foreskin surface area was assessed using Wilcoxon rank-sum tests. A sub-analysis was performed treating IL-8 as a continuous variable (when quantifiable) using Spearman's rank correlation (rho). Trends in surface area by the number of cytokines present was assessed using the Cuzick test for trend. Associations between foreskin surface area and the penile microbiome (total bacterial load and absolute abundance of specific genus) were assessed using Spearman's rank correlation. Analyses were conducted in Stata SE (Version 14.2, Revision 29, StataCorp, College Station, USA) and R (Version 3.3.1, R Development Core Team. R: A language and environment for statistical computing. *R Foundation for Statistical Computing*, Vienna, 2012).

## Results

Foreskin surface area, microbiome composition, and penile cytokine data were available from 82 men randomized to receive immediate circumcision and have been described previously [11, 12, 18, 20]. The median age was 25 years (range: 15–48 years), 46.3% of men were married (41.4% monogamous, 4.9% polygamous), median number of sexual partners in the previous year was 2 (range: 0–4 partners), and median foreskin surface area was 40.0cm$^2$ (range: 12 – 90cm$^2$; IQR 30 – 50cm$^2$).

We found no significant association between foreskin surface area and total bacterial load on the coronal sulcus (rho = 0.05; Fig 1A). We also found no associations between foreskin surface area and abundance of any of the specific anaerobic genera previously found to be associated with HIV-1 risk [11], including *Prevotella* (rho = 0.04), *Dialister* (rho = 0.12), *Peptostreptococcus* (rho = 0.10), *Mobiluncus* (rho = 0.04), *Peptoniphilus* (rho = 0.06), *Porphyromonas* (rho = 0.05), *Finegoldia* (rho = 0.07), and *Murdochiella* (rho = 0.25). We also found no significant associations between foreskin surface area and the abundance of aerobic/facultative anaerobic bacterial taxa that are common on the penis post-circumcision, including *Helococcus* (rho = 0.00), *Corynebacterium* (rho = -0.01), *Staphylococcus* (rho = 0.00).

No significant associations were observed between foreskin surface area and penile IL-8 concentration (rho = 0.05), nor the prevalence of IL-8, IL-1α, MIG, MIP-3α, RANTES, or GM-CSF. Additionally, no association was observed between foreskin surface area and the total number of cytokines detected in coronal sulcus secretions ($p_{trend}$ = 0.65; Fig 1B).

As a direct measure of the distance between coronal sulcus bacteria and the air outside the sub-preputial space, we also assessed relationships with foreskin length (where surface area is

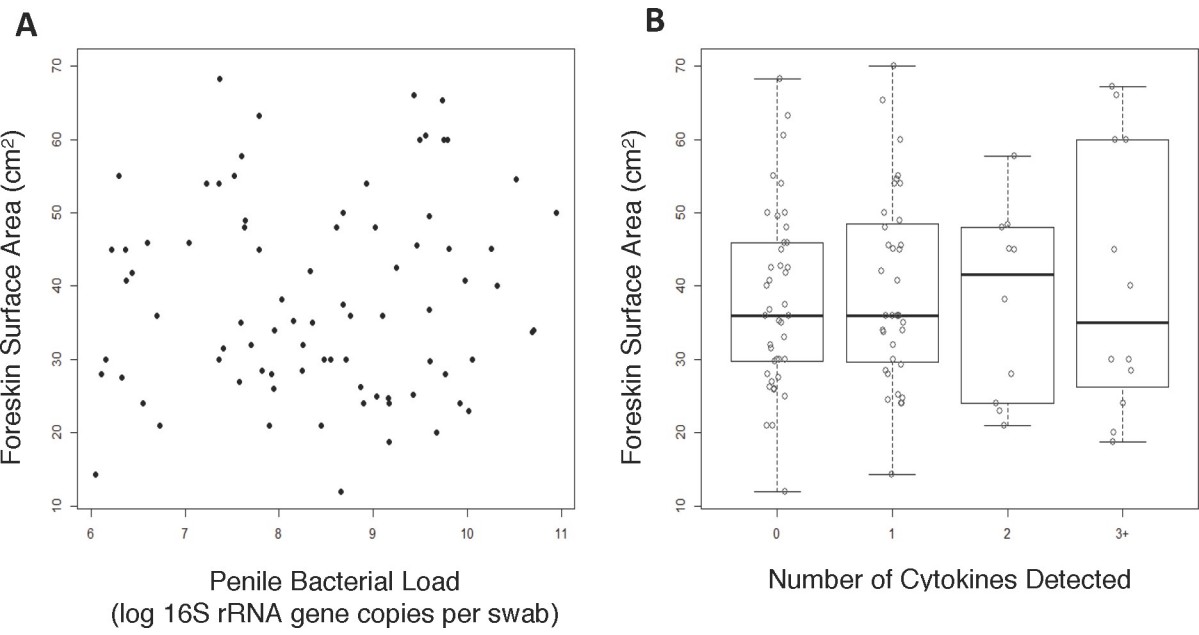

**Fig 1.** Foreskin surface area is not associated with (A) total bacterial load nor (B) number of cytokines detected. Foreskin surface area was measured on tissues removed during circumcision for HIV-1 prevention (n = 82). Penile bacterial load was estimated as 16S qPCR copy number from a swab of the coronal sulcus. Cytokines were measured in the same swab by a multiplexed enzyme-linked immunosorbent assay.

the product of length and width). We found no significant associations between foreskin length and penile cytokines or bacteria.

## Conclusions

Coverage of the glans by the foreskin on the non-erect penis is variable, and men with larger foreskin surface areas are at an increased risk of acquiring HIV [18]. We hypothesized that a larger foreskin surface area may generate a deeper fold and promote the growth of anaerobic bacteria in the sub-preputial space. Our group and others have previously demonstrated that a high abundance of genital anaerobic bacteria (such as observed in Bacterial Vaginosis in women [21]), is associated with local inflammation, immune activation, and increased risk of subsequent HIV acquisition [11, 22–28]. Genital inflammation caused by anaerobic bacteria may increase HIV risk through the recruitment of additional HIV target cells to the mucosa (HIV infects immune cells expressing the HIV co-receptors CD4 and CCR5) and by reducing epithelial barrier function through increased expression of proteases and epithelial remodeling [12, 24–26, 29]. Therefore, we hypothesized that the mechanism by which a larger foreskin surface area increases HIV risk would be through the generation of a deeper sub-preputial fold, which would specifically enhance the growth of pro-inflammatory anaerobic bacteria. To test this hypothesis, we explored the associations of foreskin size with penile microbiome composition and local inflammation.

Contrary to our hypothesis, we observed no associations between foreskin surface area and either penile microbiome composition or local levels of inflammatory cytokines. This suggests that foreskin surface area does not increase HIV risk by changing the penile microbiome composition, and presumably increases increase risk by an independent mechanism. An alternative and plausible mechanism by which a larger foreskin may increase HIV-1 risk is purely stochiometric, such that a larger foreskin surface area provides a greater number of HIV target

cells that may be exposed to HIV-containing vaginal or rectal secretions during condomless sex.

This was a retrospective analysis, and two limitations of this analysis stem from the lack of direct measurements of HIV target cell density and depth of foreskin fold at the time of the original study. In our analysis, we assumed that foreskin surface area would correlate with coverage of the glans and depth of foreskin fold, but interpersonal differences in the size of the glans penis may introduce variability in this correlation, limiting our ability to detect statistically significant associations between surface area and microbiome or cytokine outcomes. While HIV target cell density would ideally have been measured in foreskin tissues removed during circumcision, archived tissues were not available for this analysis. However, we have previously reported that IL-8 concentrations in the sub-preputial space are positively associated with tissue density of CD4+CCR5+ CD4 T cells and Th17 cells [12] (two cell types that are highly susceptible to HIV [30]).

Additional studies will be required to confirm the mechanism(s) underlying the previously described association of foreskin size with HIV-1 acquisition risk.

## Supporting information

**S1 Data.**
(XLSX)

## Acknowledgments

We would like to acknowledge the contribution of the men and women of Rakai who participated in this study.

## Author Contributions

**Conceptualization:** Godfrey Kigozi, Cindy M. Liu, Rupert Kaul, Aaron A. R. Tobian, Alison G. Abraham, Jessica L. Prodger.

**Data curation:** Godfrey Kigozi, Cindy M. Liu, Zoe R. Packman, Ronald H. Gray, Joseph Ssekasanvu, Jessica L. Prodger.

**Formal analysis:** Cindy M. Liu, Daniel Park, Zoe R. Packman, Alison G. Abraham, Jessica L. Prodger.

**Funding acquisition:** Godfrey Kigozi, Cindy M. Liu, Ronald H. Gray, Rupert Kaul, Aaron A. R. Tobian, Joseph Kagaayi, Jessica L. Prodger.

**Project administration:** Cindy M. Liu.

**Supervision:** Godfrey Kigozi, Cindy M. Liu, Ronald H. Gray, Rupert Kaul, Aaron A. R. Tobian, Alison G. Abraham, Joseph Kagaayi, Jessica L. Prodger.

**Writing – original draft:** Jessica L. Prodger.

**Writing – review & editing:** Godfrey Kigozi, Cindy M. Liu, Daniel Park, Zoe R. Packman, Ronald H. Gray, Rupert Kaul, Aaron A. R. Tobian, Alison G. Abraham, Joseph Ssekasanvu, Joseph Kagaayi, Jessica L. Prodger.

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
