## [Decision Letter · Decision Letter 0]

2 Mar 2020

PONE-D-20-00301

Foreskin surface area is not associated with sub-preputial microbiome composition or penile cytokines.

PLOS ONE

Dear Dr. Prodger,

Thank you for submitting your manuscript to PLOS ONE. After careful consideration, we feel that it has merit but does not fully meet PLOS ONE’s publication criteria as it currently stands. Therefore, we invite you to submit a revised version of the manuscript that addresses the points raised during the review process.

Specifically, Reviewer 1 has detailed a weak correlation and the suggestion of submission as a short report. Alternately, additional information in a fuller manuscript would be appropriate.

We would appreciate receiving your revised manuscript by Apr 16 2020 11:59PM. To enhance the reproducibility of your results, we recommend that if applicable you deposit your laboratory protocols in protocols.io, where a protocol can be assigned its own identifier (DOI) such that it can be cited independently in the future. For instructions see: http://journals.plos.org/plosone/s/submission-guidelines#loc-laboratory-protocols

We look forward to receiving your revised manuscript.

Kind regards,

Noreen J. Hickok, Ph.D.

Academic Editor

PLOS ONE

Journal Requirements:

Reviewers' comments:

Reviewer's Responses to Questions

**Comments to the Author**

1. Is the manuscript technically sound, and do the data support the conclusions?

Reviewer #1: Partly

Reviewer #2: Yes

2. Has the statistical analysis been performed appropriately and rigorously? 

Reviewer #1: No

Reviewer #2: Yes

3. Have the authors made all data underlying the findings in their manuscript fully available?

Reviewer #1: Yes

Reviewer #2: Yes

4. Is the manuscript presented in an intelligible fashion and written in standard English?

Reviewer #1: Yes

Reviewer #2: Yes

5. Review Comments to the Author

Reviewer #1: In this manuscript, Kigozi et al. analyzed the association between the foreskin surface area and HIV-1 risk by measuring the correlation between foreskin size and anaerobes or pro-inflammatory cytokines. While I found this work of interests, the following concerns should be raised.

1. Manuscript type. This study reports a very specific result of weak correlations between the foreskin surface area and anaerobes or pro-inflammatory cytokines. Considering the significance and the manuscript length, in this case, I would suggest the authors to format it as a “Brief report”, “Communications” or “Letter to the editor” rather than a “Research article”.

2. Technical issues. The analysis on abundance of bacteria was based-on 16S rRNA profiling, mainly on Genus-level. I strongly recommend the authors to (a) go deeper on 97% or 99% OTU level for higher precision, and (b) also from the functional aspect using 16S-based function annotation method (e.g. PICRUSt).

3. In the current results, since there is no individual genus was associated with the foreskin surface area that may influence the risk of HIV-1, is there any probability that the risk is correlated with the combination of multiple organisms? A machine-based approach / regression analysis of multiple-taxon / PCA may inspire new findings, or further confirm the current results.

4. Line #126, for specific anaerobic genera previously found to be associated with HIV-1 risk, each of them should be cited, or linked to the previous studies (e.g. listed in a table to summarize each of the genera and its citation(s)).

5. The resolution of Figure 1 should be improved.

Reviewer #2: Thank you for the opportunity to review this brief, but well-written manuscript. The study appears well-conducted and contributes novel information to the study of MMC and HIV risk. I have only minor comments:

Methods:

• It seems unusual to have ethics information at the end of the methods. If this is a journal style requirement, fine, but otherwise it is more common to lead with ethics information.

• Which R version was used? Any specific packages?

Results:

• Graphical results of numeric IL-8 analysis are mentioned in the methods but not presented.

• It’s possible that the genera previously associated with HIV risk are incorrect. Was a larger analysis conducted? I don’t necessarily advocate a fishing expedition, but given the negative results, perhaps other genera might be important here.

Discussion:

• I am not sure about the strength of the conclusions given the fact that this is a single, relatively small study. Changing “indicating” to “suggesting” on line 155 may adequately temper the language.

Supplement:

• Total bacterial load and quantitative IL8 data are not present in the spreadsheet

6. PLOS authors have the option to publish the peer review history of their article (what does this mean?). If published, this will include your full peer review and any attached files.

Reviewer #1: No

Reviewer #2: Yes: Ryan Cook

---

## [Author Response · Author response to Decision Letter 0]

21 Apr 2020

We thank the reviewers for their efforts and excellent comments. Please see below responses to specific comments.

Reviewer #1: In this manuscript, Kigozi et al. analyzed the association between the foreskin surface area and HIV-1 risk by measuring the correlation between foreskin size and anaerobes or pro-inflammatory cytokines. While I found this work of interest, the following concerns should be raised.

1. Manuscript type. This study reports a very specific result of weak correlations between the foreskin surface area and anaerobes or pro-inflammatory cytokines. Considering the significance and the manuscript length, in this case, I would suggest the authors to format it as a “Brief report”, “Communications” or “Letter to the editor” rather than a “Research article”. 

- We apologize for any lack of clarity. The goal of our report is not to imply that there are weak correlations between foreskin surface and either microbiome or immune parameters – rather, it is to demonstrate that there is no correlation. At scientific meetings, we are frequently questioned if there is a relationship between foreskin size and anaerobe burden, and this analysis is meant to address that question. We hope that the data presented are now clear and make a consistent story. Since the mandate of Plos One is to publish valid research regardless of whether the results are positive or negative, we hope that they will be amenable to publishing these negative results in full.

- With regards to manuscript format, Plos One does not offer the manuscript formats suggested by the reviewer. Plos One has a relatively broad definition for Research Articles (“reports the results of original primary research, including quantitative and qualitative studies, methods and software studies, systematic reviews, and other work”), with no word count minimum or maximum. 

2. Technical issues. The analysis on abundance of bacteria was based on 16S rRNA profiling, mainly on Genus-level. I strongly recommend the authors to (a) go deeper on 97% or 99% OTU level for higher precision, and (b) also from the functional aspect using 16S-based function annotation method (e.g. PICRUSt). 

- Thank you for this suggestion. Data are shown to the 97% OTU level, as recommended.

3. In the current results, since there is no individual genus was associated with the foreskin surface area that may influence the risk of HIV-1, is there any probability that the risk is correlated with the combination of multiple organisms? A machine-based approach / regression analysis of multiple-taxon / PCA may inspire new findings, or further confirm the current results. 

- In response to this excellent suggestion we conducted decision tree analyses to evaluate potential associations of combinations of bacterial taxa with surface area. For this analysis, we also expanded to use the full set of bacterial taxa found in the sub-preputial space, as opposed to limiting to those taxa previously associated with circumcision. Only one taxon (an OTU with 80% similarity to Gallicola) was identified as positively associated with surface area. Forcing an expansion of tree depth did not identify any informative or significant combinations of multiple organisms. The one taxon identified as being associated with surface area was only marginally informative (r=0.280), and although the association had a significant Spearman correlation, this taxon had a mean proportional abundance of less than 0.1% and is unlikely to be of biological relevance.

4. Line #126, for specific anaerobic genera previously found to be associated with HIV-1 risk, each of them should be cited, or linked to the previous studies (e.g. listed in a table to summarize each of the genera and its citation(s)). 

- The relevant citation had been added.

5. The resolution of Figure 1 should be improved. 

- This figure has been uploaded as a high-resolution tiff, and we hope that the reviewer is now able to view it in a high res format.

Reviewer #2: Thank you for the opportunity to review this brief, but well-written manuscript. The study appears well-conducted and contributes novel information to the study of MMC and HIV risk. I have only minor comments.

- Thank you for the kind comments.

Methods:

• It seems unusual to have ethics information at the end of the methods. If this is a journal style requirement, fine, but otherwise it is more common to lead with ethics information.

- These have now been placed at the start of the Methods, as suggested.

• Which R version was used? Any specific packages?

- The R version used was R Development Core Team, Version 3.3.1. This information has now been added to the Methods section (lines 116-118).

Results:

• Graphical results of numeric IL-8 analysis are mentioned in the methods but not presented.

- Thank you, the methods have been corrected (line 112). 

• It’s possible that the genera previously associated with HIV risk are incorrect. Was a larger analysis conducted? I don’t necessarily advocate a fishing expedition, but given the negative results, perhaps other genera might be important here.

- Our prior publication demonstrated eight genera associated with HIV risk in uncircumcised men, but the reviewer is correct that the analysis was limited to genera that had previously demonstrated to be reduced by circumcision. To determine if other taxa are associated with surface area, we performed additional analyses expanding to use the full set of bacterial taxa found in the sub-preputial space. Only one taxon (an OTU with 80% similarity to Gallicola) was identified as being marginally informative in predicting surface area (positive association, r=0.280). Although the association had a significant Spearman correlation, this taxon had a mean proportional abundance of less than 0.1% and is therefore unlikely to be of biological relevance.

Discussion:

• I am not sure about the strength of the conclusions given the fact that this is a single, relatively small study. Changing “indicating” to “suggesting” on line 155 may adequately temper the language.

- We have tempered the wording, as suggested. 

Supplement:

• Total bacterial load and quantitative IL8 data are not present in the spreadsheet

- Our apologies for the oversight, those data have now been added.

---

## [Editor Report · Decision Letter 1]

24 Apr 2020

PONE-D-20-00301R1

Foreskin surface area is not associated with sub-preputial microbiome composition or penile cytokines.

PLOS ONE

Dear Dr. Prodger,

Thank you for submitting your manuscript to PLOS ONE. After careful consideration, we feel that it has merit but does not fully meet PLOS ONE’s publication criteria as it currently stands. Therefore, we invite you to submit a revised version of the manuscript that addresses the points raised during the review process.

Specifically, I have asked you to expand your discussion to place it more securely in the context of the field, especially your previous work.

We would appreciate receiving your revised manuscript by Jun 08 2020 11:59PM. To enhance the reproducibility of your results, we recommend that if applicable you deposit your laboratory protocols in protocols.io, where a protocol can be assigned its own identifier (DOI) such that it can be cited independently in the future. For instructions see: http://journals.plos.org/plosone/s/submission-guidelines#loc-laboratory-protocols

We look forward to receiving your revised manuscript.

Kind regards,

Noreen J. Hickok, Ph.D.

Academic Editor

PLOS ONE

Additional Editor Comments (if provided):

These findings are of significance because they (1) show no correlation between foreskin size and anaerobic bacterial load and (2) because they are on update on your previous findings that DID suggest a correlation. Could you please address your previous studies and succinctly point out the assumptions/analyses/small sample sizes that led you to draw your previous conclusions...and how they differ from your current study. Also, please address the issue that these are all adults who are being circumcised to lower their chances of contracting AIDS yet they don't have AIDS and they presumably have been sexually active. Is it possible that these subjects are also skewed? Thus, please bring the paper into context on the level of subject selection and previous studies.

---

## [Author Response · Author response to Decision Letter 1]

27 Apr 2020

Dear Dr. Hickok 

Our group has previously published two independent analyses using data collected during the randomized controlled trial of circumcision in Uganda: one showing an association between foreskin surface area and risk of HIV acquisition (Aids 2009), and a second analysis showing associations between penile anaerobes, pro-inflammatory cytokines, and HIV risk (MBio 2017). However, our group (and, to our knowledge, no other group) has previously assessed the relationship between foreskin surface area and penile anaerobes and/or pro-inflammatory cytokines. Therefore, the analyses presented in the current manuscript are not an update on any previous findings. I apologize for this lack of clarity, and have altered the wording in the introduction in lines 79-80 to try to address this issue.

Another concern raised was the possibility of selection bias for enrollment of men who had been exposed to HIV but remained uninfected. At the time of enrollment in the trial there was no evidence that circumcision reduced HIV risk, and thus enrollment was performed at a community level without selection based on sexual activity or prior HIV exposure. Additionally, the present analyses were performed on samples collected at trial initiation, and thus no additional selection bias was introduced by a requirement of remaining HIV negative throughout the trial period. This has now been clarified on lines 93-95.

I hope that we have adequately addressed your two remaining concerns, and have uploaded a new version of the manuscript with these changes tracked. Thank you for considering our manuscript for publication in Plos One.

Sincerely,

Jessica Prodger

---

## [Editor Report · Decision Letter 2]

29 Apr 2020

PONE-D-20-00301R2

Foreskin surface area is not associated with sub-preputial microbiome composition or penile cytokines.

PLOS ONE

Dear Dr. Prodger,

Thank you for submitting your manuscript to PLOS ONE. After careful consideration, we feel that it has merit but does not fully meet PLOS ONE’s publication criteria as it currently stands. Therefore, we invite you to submit a revised version of the manuscript that addresses the points raised during the review process.

I am afraid that I am asking you to look at your discussion again and to expand this section as detailed below.

We would appreciate receiving your revised manuscript by Jun 13 2020 11:59PM. To enhance the reproducibility of your results, we recommend that if applicable you deposit your laboratory protocols in protocols.io, where a protocol can be assigned its own identifier (DOI) such that it can be cited independently in the future. For instructions see: http://journals.plos.org/plosone/s/submission-guidelines#loc-laboratory-protocols

We look forward to receiving your revised manuscript.

Kind regards,

Noreen J. Hickok, Ph.D.

Academic Editor

PLOS ONE

Additional Editor Comments (if provided):

I am sorry, but I still ask you to expand your Conclusions/Discussion section. In your introduction, you state,

"We previously reported that on the penis both the density of specific anaerobic bacterial genera

67 and local levels of pro-inflammatory cytokines are associated with increased risk of HIV-1

68 seroconversion in uncircumcised men [5, 6]. Additionally, the abundance of anaerobic genera

69 on the penis correlates positively with local concentrations of these cytokines [5]. One possible

70 interpretation for these findings is that anaerobic bacteria increase HIV-1 risk by driving penile

71 inflammation. Mucosal inflammation is a demonstrated risk factor for cervico-vaginal HIV-1

72 acquisition, and is associated with an increased number and relative susceptibility of HIV-1

73 target cells in the mucosa [7, 8], altered dendritic cell sampling [9], and decreased epithelial

74 barrier function [10, 11].

75 An epidemiologic study showed that a larger foreskin surface area is also associated with an

76 increased risk of HIV-1 seroconversion [12]. Since coverage of the glans on the non-erect penis

77 is variable and dependent on foreskin size, a deeper foreskin fold from a larger foreskin could

78 result in a less aerobic environment that might preferentially promote anaerobic bacterial growth

79 and induce inflammation. However, it has not yet been explored if foreskin size is associated

80 with penile microbiome composition and local inflammation."

I ask that you indicate any weaknesses in your study as well as putting these studies into the context of the other studies that you have cited. Some explanation of what they found and what you found, other than it is not what you found, is needed. Please expand this section so as to increase its relation to the publications in the field.
---

## [Author Response · Author response to Decision Letter 2]

21 May 2020

We have significantly expanded the discussion, and hopefully have now placed our findings within the context of previous publications in the field (lines 156 – 170). We have also included a section detailing what we believe to be the two most important limitations of our study: the lack of direct measurements of (i) depth of foreskin fold, and (ii) HIV target cell density (lines 182 – 192). To our knowledge, no other publications have previously investigated the relationship between foreskin size (or fold depth) and penile microbiome composition or penile immune outcomes.

---

## [Editor Report · Decision Letter 3]

22 May 2020

Foreskin surface area is not associated with sub-preputial microbiome composition or penile cytokines.

PONE-D-20-00301R3

Dear Dr. Prodger,

We are pleased to inform you that your manuscript has been judged scientifically suitable for publication and will be formally accepted for publication once it complies with all outstanding technical requirements.

With kind regards,

Noreen J. Hickok, Ph.D.

Academic Editor

PLOS ONE
---

## [Editor Report · Acceptance letter]

12 Jun 2020

PONE-D-20-00301R3 

Foreskin surface area is not associated with sub-preputial microbiome composition or penile cytokines. 

Dear Dr. Prodger:

I'm pleased to inform you that your manuscript has been deemed suitable for publication in PLOS ONE. Congratulations! Your manuscript is now with our production department. 

Kind regards, 

on behalf of

Dr. Noreen J. Hickok 

Academic Editor

PLOS ONE